# Comparison of the Oncological Outcomes of Open versus Laparoscopic Surgery for T2 Gallbladder Cancer: A Propensity-Score-Matched Analysis

**DOI:** 10.3390/jcm11092644

**Published:** 2022-05-08

**Authors:** Jin-Kyu Cho, Jae-Ri Kim, Jae-Yool Jang, Han-Gil Kim, Jae-Myung Kim, Seung-Jin Kwag, Ji-Ho Park, Ju-Yeon Kim, Young-Tae Ju, Chi-Young Jeong

**Affiliations:** 1Department of Surgery, Gyeongsang National University Hospital, Gyeongsang National University College of Medicine, 79, Gangnam-ro, Jinju 52727, Korea; hbpcjk@naver.com (J.-K.C.); drkhg@naver.com (H.-G.K.); jmjidia@hanmail.net (J.-M.K.); drksj77@naver.com (S.-J.K.); goodgsdr@gmail.com (J.-H.P.); juyeon0910@hanmail.net (J.-Y.K.); drjyt@naver.com (Y.-T.J.); 2Department of Surgery, Gyeongsang National University Changwon Hospital, Gyeongsang National University College of Medicine, 11, Samjeongja-ro, Changwon-si 51472, Korea; jaripo@gmail.com (J.-R.K.); alitaalita@naver.com (J.-Y.J.)

**Keywords:** gallbladder carcinoma, oncological outcome, laparoscopy

## Abstract

Although laparoscopic treatment for T1 gallbladder cancer (GBC) has been described previously, the differences in oncologic outcomes between laparoscopic and conventional open surgery for T2 GBC have not been investigated. We aimed to assess the role of laparoscopic surgery using retrospectively collected data for 81 patients with T2 GBC who underwent surgical resection between January 2010 and December 2017. Eligible patients were classified into “laparoscopic” and “open” groups. Propensity-score matching was performed in a 1:1 ratio. The effects of surgery type on surgical and oncological outcomes were investigated. After propensity-score matching, 19 patients were included in the open and laparoscopic surgery groups. The median follow-up durations were 70 and 26 months in the open and laparoscopic groups, respectively. The operative time (316.8 ± 80.3 vs. 218.9 ± 145.0 min, *p* = 0.016) and length of postoperative hospital stay (14.4 ± 6.0 vs. 8.4 ± 5.9 days, *p* = 0.004) were significantly shorter in the laparoscopic group. The three-year overall (86.3% vs. 88.9%, *p* = 0.660) and disease-free (76.4% vs. 60.2%, *p* = 0.448) survival rates were similar between the groups. Propensity-score matching showed that laparoscopic surgery for T2 GBC yielded similar long-term oncological outcomes and favorable short-term outcomes in comparison with open surgery. Laparoscopic treatment should be considered in patients with T2 GBC.

## 1. Introduction

Gallbladder cancer (GBC) is the fifth most common carcinoma of the gastrointestinal tract and the most common carcinoma of the biliary tract [1], with an overall incidence of 3 per 100,000 persons [2]. Curative resection is the only effective treatment for GBC [3,4]. Conventional open extended cholecystectomy, including dissection of the regional lymph node (LN) and wedge resection of the gallbladder bed, is the standard curative resection technique for GBC [5,6].

Laparoscopic surgery was originally associated with a risk of inadequate curative resection and tumor cell spread during surgery [3,4,7]. However, with advancements in laparoscopic instruments and accumulation of surgical skills, laparoscopic surgical treatment has gained acceptance as a standard treatment method with oncological outcomes comparable to those of conventional open treatment. Moreover, laparoscopic surgery is widely used for various cancers, including stomach, colon, and rectal cancer [8,9,10,11]. With the increased use of laparoscopic techniques for gallbladder disease treatment, cases of incidentally discovered GBC after laparoscopic surgery are steadily increasing [12,13]. Additionally, with an increase in the number of laparoscopic approaches for GBC, the oncological adequacy of laparoscopic surgery, specifically for patients with T2 GBC, has become an important topic of debate [14].

Several recent studies have reported the feasibility of laparoscopic treatment for GBC [14,15,16,17,18]. However, these studies were retrospective and nonrandomized, and included only small numbers of cases [15,16,17,18]. Therefore, their findings were debatable, and the use of laparoscopic treatment for T2 GBC remains controversial, with oncological outcomes being challenging to determine.

Retrospective studies can analyze large sample sizes but excluding selection and severity biases in such studies can be difficult. Using a propensity score (PS) to compare the two groups can reduce such biases by adjusting the observed pretreatment characteristics [19]. Accordingly, we used PS-matching analysis to evaluate and compare the feasibility and oncological outcomes of laparoscopic and open surgeries for T2 GBC.

## 2. Materials and Methods

The need to obtain informed consent from participants was waived owing to the retrospective nature of the study, and the study was conducted in accordance with the Declaration of Helsinki. The institutional review board of Gyeongsang National University Hospital approved this retrospective study (approval number: GNUH 2017-03-018).

### 2.1. Patient Selection

We retrospectively analyzed the medical data of patients who underwent laparoscopic or open surgery for GBC at Gyeongsang National University Hospital from January 2010 to December 2017. Patients with pathologically proven T2 GBC who underwent curative resection were included. The exclusion criteria were (1) a history of another primary malignancy, (2) incomplete resection, (3) combined resection with other organs, or (4) incomplete medical records. Patients were classified into laparoscopic and open surgery groups according to the type of surgery they underwent. Patients who underwent open surgery after laparoscopic surgery were considered as having undergone open surgery.

### 2.2. Surgical Procedure for GBC

The surgical procedure selected for GBC was based on the recommendations of the Korean Association of Hepatobiliary and Pancreas Surgery: Simple cholecystectomy for T1a GBC, simple or extended cholecystectomy for T1b GBC, and extended cholecystectomy for T2 GBC or above. The use of cholecystectomy alone was defined as simple cholecystectomy. Cholecystectomy with further resection included LN dissection, liver resection, and/or bile duct resection. Patients who refused additional extended resection after simple cholecystectomy were followed-up without further intervention.

At our institution, laparoscopic surgery is recommended for cases of suspected T1 or T2 GBC identified by preoperative abdominal computed tomography (CT) (no liver infiltration and no involvement of extrahepatic adjacent organs), based on the 26th World Congress of the International Association of Surgeons, Gastroenterologists, and Oncologists expert consensus [20]. Open surgery is recommended for patients showing extensive liver infiltration on CT, extrahepatic bile duct or adjacent organ involvement, and incidental diagnosis of GBC after open cholecystectomy. Open surgery is also recommended for patients who refuse to undergo laparoscopic treatment.

### 2.3. PS-Matching Analysis

To achieve balance in the baseline variables between the laparoscopic and open surgery groups, PS matching was performed [21]. Patients in the laparoscopic surgery group were PS-matched in a 1:1 ratio with patients in the open surgery group. Many propensity models were tested with various covariates such as age, gender, preoperative American Society of Anesthesiologists (ASA) score, combined GB stone, pathologic T stage, pathologic N stage, simple cholecystectomy versus cholecystectomy with further resection, elevated carcinoembryonic antigen, elevated carbohydrate antigen 19-9, and adjuvant chemotherapy. After propensity score matching, we calculated the C-statistic and the standardized difference to get the best model. We chose the best-balanced PS matching model, which contained five covariates: Age, preoperative ASA score, pathologic T stage, simple cholecystectomy versus cholecystectomy with further resection, and adjuvant chemotherapy. After matching, all covariates had reduced standardized differences and were well balanced between the two groups (C-statistic = 0.808).

### 2.4. Measurements

The following patient data were collected: Age, sex, body mass index, ASA score, presence of combined gallbladder stones, laboratory findings, preoperative tumor markers (carcinoembryonic antigen [CEA] and carbohydrate antigen 19-9), operative time, Clavien–Dindo classification [22,23], pathologic tumor size, pathologic tumor stage, pathologic LN stage, number of metastatic LNs, number of retrieved LNs, adjuvant chemotherapy, postoperative hospital stay, recurrence site, date of recurrence, and date of death. Pathological TN stage was defined according to the 8th edition of the American Joint Committee on Cancer. We compared the disease-free survival (DFS) and cancer-specific overall survival (OS) rates between the two groups. DFS was defined as the time from diagnosis to first recurrence. OS was defined as the time from diagnosis to death owing to a cancer-specific cause.

### 2.5. Statistical Analysis

Statistical analysis was performed using SPSS version 22.0 (Released 2013; IBM Corp., Armonk, NY, USA). Statistical significance was set at *p* < 0.05. Categorical variables were expressed as the number of cases and percentage (%), and the chi-square test or Fisher’s exact test was used for univariate analysis. Univariate and multivariate analyses were performed using the Cox proportional hazards model to identify the factors associated with survival. The risks were expressed as hazard ratios (HRs) and 95% confidence intervals (Cis). CA19-9 and CEA were considered elevated if they were greater than 37 U/mL and greater than 5 ng/mL, respectively, according to the laboratory cut-off values often used in our center. Previous studies have routinely employed the cut-off age and tumor size of 60 years and greater than 1 cm. The Kaplan–Meier method was used to analyze survival, and variables were examined using a log-rank test.

## 3. Results

### 3.1. Patient Characteristics

Between January 2010 and December 2017, 92 patients were diagnosed with T2 GBC and underwent surgical treatment. Of these, 11 were excluded from the analysis because of a history of another primary malignancy (*n* = 4), incomplete resection (*n* = 3), combined resection of other organs (*n* = 3), or incomplete medical records (*n* = 1). Ultimately, 81 patients were included in the analysis.

### 3.2. Before PS Matching

Data on surgery type, baseline characteristics, and short-term surgical outcomes in the laparoscopic and open surgery groups before PS matching are shown in Table 1, Table 2 and Table 3, respectively. Laparoscopic and open surgeries were performed in 37 and 44 patients, respectively. The two groups showed no significant differences in surgery types. Simple cholecystectomy and cholecystectomy with LN dissection were predominant in the laparoscopic surgery group, whereas cholecystectomy with LN dissection and liver resection were predominant in the open surgery group (Table 1). Significant differences were observed between the groups in terms of age and ASA scores (Table 2). The surgical outcomes also differed between the laparoscopic and open surgery groups. The operative time was significantly shorter in the laparoscopic group than in the open surgery group (165.8 ± 128.8 vs. 332.3 ± 93.3 min, *p* < 0.001). The open surgery group had more retrieved LNs, had more metastatic LNs, showed a more advanced N stage, and had longer hospital stays than the laparoscopic surgery group (Table 3). The median follow-up durations in the laparoscopic and open surgery groups were 21 and 48 months, respectively. The Kaplan–Meier curves for DFS and OS are shown in Figure 1 and Figure 2, respectively. The two groups showed no significant differences in the three-year DFS and OS rates (DFS: 65.0% vs. 66.7%, *p* = 0.721; cancer-specific OS: 78.0% vs. 82.4%, *p* = 0.782).

### 3.3. Comparison of the Laparoscopic and Open Surgery Groups after PS Matching

Nineteen patients in each group were selected for PS-matching analyses. All covariates were well-balanced between the two groups (C-statistic = 0.808). The surgery types were matched between the two groups (Table 4). The baseline characteristics of the PS-matched patients are presented in Table 2. Preoperative conditions, laboratory findings, and tumor markers were not significantly different between the two groups. However, with regard to surgical outcomes, the laparoscopic group had a significantly shorter operative time and hospital stay than the open surgery group (218.9 ± 145.0 vs. 316.8 ± 80.3 min, *p* = 0.016; 8.4 ± 5.9 vs. 14.4 ± 6.0 days, *p* = 0.004, respectively; Table 3). No significant differences were observed in the other clinicopathological factors, including metastatic LNs (0.4 ± 0.9 vs. 0.4 ± 1.0, *p* > 0.999), retrieved LNs (5.3 ± 6.3 vs. 7.3 ± 5.5, *p* = 0.319), and complication rate (21.1% vs. 10.5%, *p* = 0.660). Postoperative complications occurred in four (21.1%) patients in the laparoscopic surgery group, including wound infection (*n* = 3) and bile leakage (*n* = 1). In comparison, two (10.5%) patients in the open surgery group had postoperative complications, including symptomatic fluid collection in the gallbladder bed (*n* = 1) and wound infection (*n* = 1). No gallbladder perforation occurred during surgery, and no deaths were observed in either group. The median follow-up durations in the laparoscopic and open surgery groups were 26 and 70 months, respectively. The Kaplan–Meier curves for DFS and OS are shown in Figure 3 and Figure 4, respectively. The two groups showed no significant differences in the three-year DFS and OS rates (DFS: 60.2% vs. 76.4%, *p* = 0.448; cancer-specific OS: 88.9% vs. 86.3%, *p* = 0.660).

### 3.4. Prognostic Factors for T2 GBC

To identify the prognostic factors of survival in T1–T2 GBC, we used univariate Cox regression analysis, and the results are presented in Table 5. In the univariate analysis, LN metastasis and elevated CEA levels (>5 ng/mL) were significantly associated with poorer oncologic outcomes in T2 GBC. In the multivariate Cox regression analysis, LN metastasis was independently associated with T2 GBC survival (HR = 9.336, 95% CI = 2.295–37.985, *p* = 0.002). The type of surgery (laparoscopic vs. open surgery) was not a prognostic factor (HR = 1.130, 95% CI = 0.247–5.167, *p* = 0.875).

## 4. Discussion

This study compared the surgical outcomes associated with laparoscopic and open surgery using PS-matching analysis for T2 GBC. Our results showed that the effectiveness of laparoscopic surgery for T2 GBC was not inferior to that of open surgery in terms of perioperative outcomes and the three-year DFS and OS rates. Additionally, laparoscopic surgery offers significant functional advantages, such as a shorter operative time and length of hospital stay.

Improvements in instrumentation and advanced surgical skills have led to the widespread use of laparoscopic treatment for gastrointestinal tract cancers [24]. Laparoscopic treatment has been accepted as a standard method for early-stage tumors, with oncological and surgical outcomes comparable to those of open surgery [8,9,10,11]. Several studies have reported that laparoscopic surgery for patients with T1 GBC leads to similar or better treatment outcomes than open surgery [17,24,25,26]. Recently, laparoscopic surgery has become feasible at selected high-volume referral centers and has shown outcomes similar to those of open surgery in patients with T2 GBC [15,16,27,28,29]. However, the clinical value of laparoscopic surgery for T2 GBC remains controversial, and current guidelines such as those by the National Comprehensive Cancer Network and the Japanese Society of Hepato-Biliary-Pancreatic Surgery do not recommend laparoscopic surgery for T1 and T2 GBC because it is associated with a higher risk of tumor dissemination and port-site recurrence than open surgery [3,5,6,30]. Port-site recurrence and tumor dissemination due to gallbladder perforation have been observed after laparoscopic cholecystectomy, even in patients with early-stage GBC [7,31]. However, these reports were based on older studies, and gallbladder perforation occurred predominantly in patients with suspected benign pathology, with dissection in the thin cystic plate [29]. Moreover, tumor dissemination is not a specific complication of laparoscopic surgery and can also occur in open surgery. Appropriate use of a plastic endo-bag and careful management of the gallbladder can prevent port-site recurrence and tumor dissemination [14,20].

In this study, we found that laparoscopy was associated with oncological outcomes comparable to those in open treatment, with the additional advantages of shorter operative time and length of hospital stay. Similarly, Cho et al. [17], Gumbs et al. [28], and Agarwal et al. [29] reported that radical laparoscopic surgery is a feasible treatment modality with oncologic outcomes comparable to those of open surgery. However, these studies had several limitations, including the use of a non-randomized design, which may have led to a selection bias between the laparoscopic and open surgery groups [15,16,17,18,29]. Consequently, we used PS matching to reduce the possibility of selection bias and obtain high-quality evidence. To the best of our knowledge, the present study is the first to compare the surgical and long-term oncological outcomes of laparoscopic and open surgery for T2 GBC using PS-matching analysis.

LN dissection is necessary for curative resection because LN metastasis is an independent prognostic factor for GBC [32], and LN metastasis occurs at a rate of up to 62% in GBC [33,34,35,36,37]. This study reported that the only independent prognostic factor affecting oncological outcomes in T2 GBC was LN metastasis. The surgical approach did not affect the oncological outcomes.

However, this study has some limitations. First, only a small number of patients were included in this study. Second, the retrospective single-center design of the study may limit the generalizability of the results. Larger prospective studies are required to verify our findings.

## 5. Conclusions

In conclusion, laparoscopic surgery could become the standard treatment modality for T2 GBC patients owing to its favorable short-term and long-term outcomes.

## Figures and Tables

**Figure 1 jcm-11-02644-f001:**
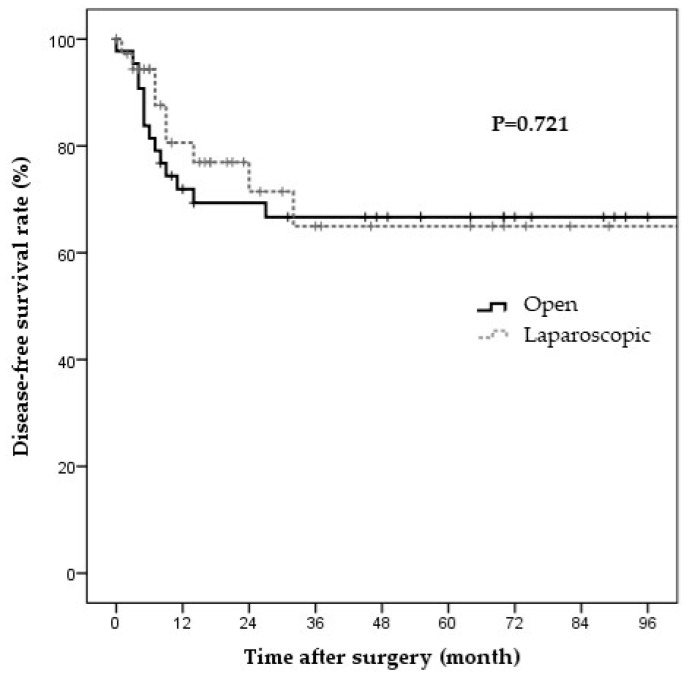
Disease-free survival in the T2 GBC patients by surgery type. The disease-free survival rate in the propensity-score-matched patients at three years was 65.0% and 66.7% in the laparoscopic and open surgery groups, respectively. The two groups showed no significant difference in recurrence (*p* = 0.721).

**Figure 2 jcm-11-02644-f002:**
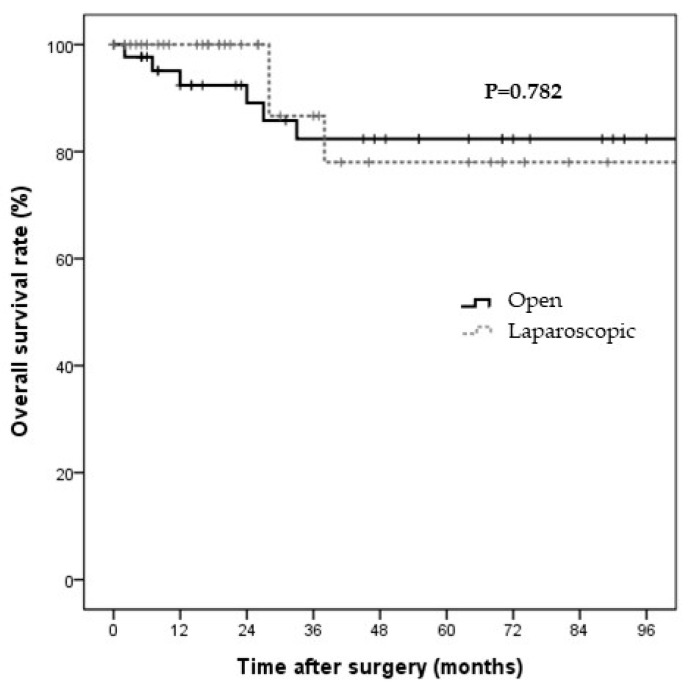
The cancer-specific survival rate in T2GBC patients by surgery type. The cancer-specific survival rates in the propensity-score-matched patients at three years were 78.0% and 82.4% in the laparoscopic and open surgery groups, respectively. No significant differences were observed between the laparoscopic and open surgery groups (*p* = 0.782) in terms of survival.

**Figure 3 jcm-11-02644-f003:**
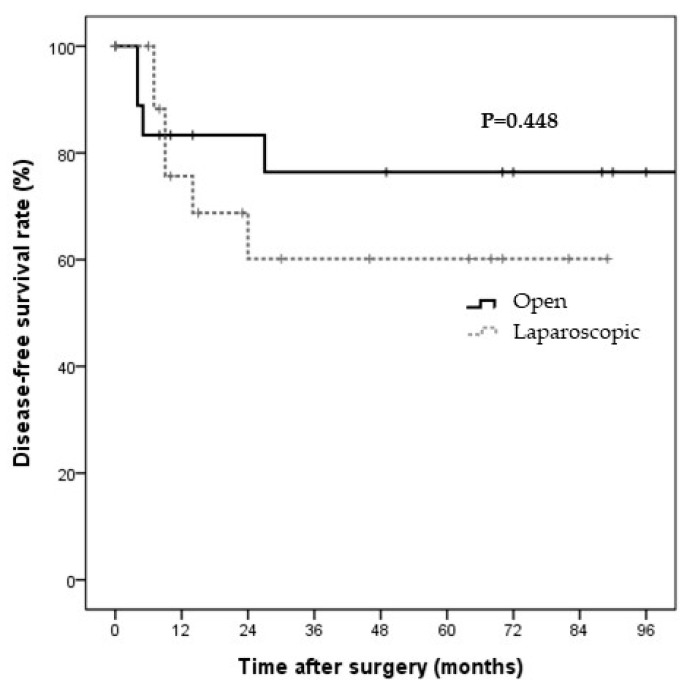
Disease-free survival in the propensity-score-matched patients by surgery type. The disease-free survival rate in the propensity-score-matched patients at three years was 60.2% and 76.4% in the laparoscopic and open surgery groups, respectively. The two groups showed no significant difference in recurrence (*p* = 0.448).

**Figure 4 jcm-11-02644-f004:**
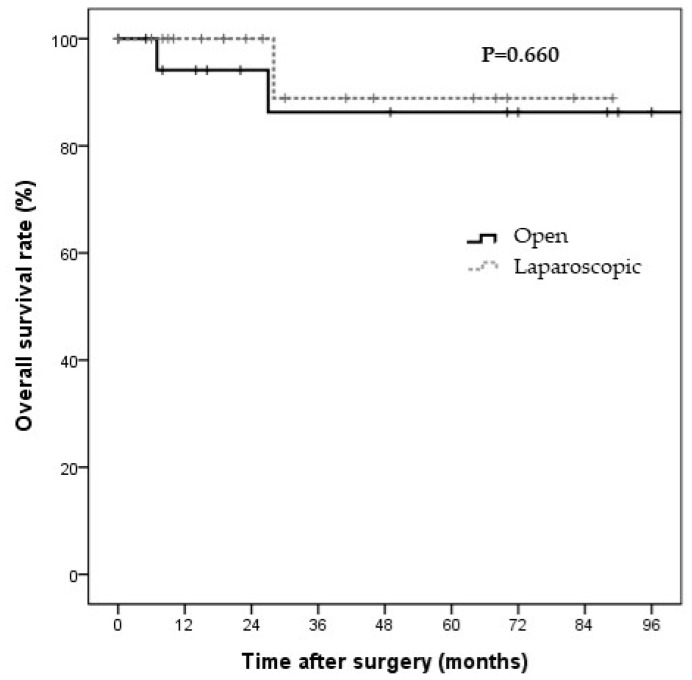
The cancer-specific survival rate in the propensity-score-matched patients by surgery type. The cancer-specific survival rates in the propensity-score-matched patients at three years were 88.9% and 86.3% in the laparoscopic and open surgery groups, respectively. No significant differences were observed between the laparoscopic and open surgery groups (*p* = 0.660) in terms of survival.

**Table 1 jcm-11-02644-t001:** Comparison of surgery type in patients before propensity-score matching.

Type of Surgery	Laparoscopic (*n* = 37)	Open (*n* = 44)	*p*-Value
Simple cholecystectomy	18 (48.6%)	5 (11.4%)	<0.001
Cholecystectomy + LND	11 (29.7%)	2 (4.5%)	
Cholecystectomy + LND + HR	7 (18.9%)	31 (70.5%)	
Cholecystectomy + LND + BDR	0 (0%)	1 (2.3%)	
Cholecystectomy + LND + HR + BDR	1 (2.7%)	5 (11.4%)	

LND, dissection of regional lymph nodes; HR, gallbladder bed wedge resection; BDR, common bile duct resection.

**Table 2 jcm-11-02644-t002:** Baseline characteristics of the patients.

	Before PS Matching			PS-Matched		
Variable	Lap. (*n* = 37) *	Open (*n* = 44) *	*p*-Value	Lap. (*n* = 19) *	Open (*n* = 19) *	*p*-Value
Age (years)	72.1 ± 9.3	63.7 ± 9.6	<0.001	69.9 ± 9.1	66.7 ± 7.8	0.251
Sex (M:F)	16:21	26:18	0.184	8:11	12:7	0.330
BMI (kg/m^2^)	23 ± 3.1	22.6 ± 2.9	0.591	22.9 ± 3.1	23.0 ± 3.1	0.933
ASA (1/2/3/4)	1/16/18/1	2/33/9/0	0.009	0/14/5/0	0/14/5/0	>0.999
Combined GB stone	11 (29.7%)	5 (11.4%)	0.051	4 (21.1%)	1 (5.3%)	0.340
Total bilirubin (mg/dL)	1.2 ± 1.8	1.2 ± 1.2	0.924	1.6 ± 2.4	1.4 ± 1.4	0.794
AST (U/L)	23.6 ± 18	41.1 ± 70.2	0.144	21.4 ± 17.7	27.1 ± 22.7	0.400
ALT (U/L)	22.2 ± 14.9	47.7 ± 73.8	0.042	22.1 ± 14.5	30.0 ± 25.4	0.253
CEA (ng/mL)	4.5 ± 7.7	4.1 ± 6.4	0.827	3.3 ± 4.0	1.81 ± 0.6	0.152
CA19-9 (U/mL)	44 ± 87.3	55.3 ± 108.9	0.657	33.7 ± 32.2	38.1 ± 37.0	0.747

* Categorical variables are expressed as percentages and continuous variables are expressed as mean ± standard deviation. PS, propensity score; Lap, laparoscopic; BMI, body mass index; ASA, American Society of Anesthesiologists physical status classification system; GB, gallbladder; AST, aspartate aminotransferase; ALT, alanine aminotransferase; CEA, carcinoembryonic antigen; CA19-9, carbohydrate antigen 19-9.

**Table 3 jcm-11-02644-t003:** Surgical outcomes of the patients.

	Before PS Matching			After PS Matching		
Variable	Lap. (*n* = 37) *	Open (*n* = 44) *	*p*-Value	Lap. (*n* = 21) *	Open (*n* = 21) *	*p*-Value
Operative time (min)	165.8 ± 128.8	332.3 ± 93.3	<0.001	218.9 ± 145.0	316.8 ± 80.3	0.016
Complication rate	5 (13.5%)	11 (25%)	0.265	4 (21.1%)	2 (10.5%)	0.660
Clavien–Dindo classification (I, II, IIIa/IIIb, IV, V)	5/0 (13.5%/0%)	10/1 (22.7%/2.3%)	0.392	4/0 (21.1%/0%)	2/0 (10.5%/0%)	0.660
Tumor size (mm)	24.6 ± 14	31.8 ± 18.6	0.052	23.1 ± 11.2	27.4 ± 15.6	0.341
T2a	15 (40.5%)	21 (47.7%)	0.654	7 (33.3%)	7 (33.3%)	>0.999
T2b	22 (59.5%)	23 (52.3%)		12 (57.1%)	12 (57.1%)	
N0	16 (43.2%)	27 (61.4%)	<0.001	12 (63.2%)	14 (73.7%)	0.693
N1	6 (16.2%)	14 (31.8%)		4 (21.1%)	4 (21.1%)	
N2	0 (0%)	2 (4.5%)		0 (0%)	0 (0%)	
Nx	15 (40.5%)	1 (2.3%)		3 (15.8%)	1 (5.3%)	
No. of positive LNs	0.3 ± 0.7	1.1 ± 2.1	0.024	0.4 ± 0.9	0.4 ± 1.0	>0.999
No. of retrieved LNs	3.4 ± 5.4	8.2 ± 5.2	<0.001	5.3 ± 6.6	7.3 ± 5.5	0.319
Adjuvant chemotherapy	11 (29.7%)	22 (50%)	0.074	7 (36.8%)	7 (36.8%)	>0.999
Length of hospital stay (day)	6.8 ± 4.9	15 ± 7.5	<0.001	8.4 ± 5.9	14.4 ± 6.0	0.004

* Categorical variables are expressed as percentages and continuous variables are expressed as mean ± standard deviation. PS, propensity score; Lap, laparoscopic.

**Table 4 jcm-11-02644-t004:** Comparison of surgery type in the propensity-score-matched patients.

Type of Surgery	Laparoscopic (*n* = 19)	Open (*n* = 19)
Simple cholecystectomy	4 (21.1%)	4 (21.1%)
Cholecystectomy with further resection	15 (78.9%)	15 (78.9%)

**Table 5 jcm-11-02644-t005:** Prognostic factors in T2 gallbladder cancer patients (*n* = 81).

Variables	Univariate Analysis			Multivariate Analysis		
	HR	95% CI	*p*-Value	HR	95% CI	*p*-Value
Female sex	1.403	0.376–5.232	0.514			
Age >60 years	1.489	0.308–7.205	0.621			
Overweight (BMI > 25 kg/m^2^)	1.632	0.437–6.093	0.466			
CEA (>5 ng/mL)	6.328	1.134–35.320	0.035	3.608	0.556–23.395	0.179
CA19-9 (>37 U/mL)	26.762	0.010–68,340.593	0.412			
Further resection	0.700	0.175–2.801	0.614			
GB stone	1.625	0.203–13.005	0.647			
Tumor size (>1 cm)	24.395	0.003–202,140.970	0.488			
T stage (T2a vs. T2b)	6.515	0.814–52.138	0.077			
Node metastasis	9.336	2.295–37.985	0.002	9.336	2.295–37.985	0.002
Complication	1.467	0.303–7.095	0.634			
Adjuvant chemotherapy	1.717	0.426–6.925	0.447			
Laparoscopic surgery	0.822	0.204–3.309	0.783	1.130	0.247–5.167	0.875

HR, hazard ratio; CI, confidence interval; BMI, body mass index; CEA, carcinoembryonic antigen; CA19-9, carbohydrate antigen 19-9; GB, gallbladder.

## Data Availability

The data presented in this study are available on request from the corresponding author.

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
