# Peer review of "Comparison of the Oncological Outcomes of Open versus Laparoscopic Surgery for T2 Gallbladder Cancer: A Propensity-Score-Matched Analysis"

_jcm, 2022, doi:10.3390/jcm11092644_

Round 1

Reviewer 1 Report

Comments for the Authors:

The manuscript titled “Comparison of the oncological outcomes of open vs laparoscopic surgery for T1-T2 gallbladder cancer: A propensity score-matched analysis” report a single center series of over 100 resected early stage gallbladder cancers. The topic is particularly interesting and discussed in the HPB scientific community.

However, I have some concerns about the methodology used. In fact, the will to complete the study with a PSM, reduces the number and quality of the groups compared without adding important data but opening some doubts:

  • Looking at the matched cohort I noted that there are 2 T1a and 19 T2 but no T1b. So, considering that for T1a GBC a simple cholecystectomy is recommended, the real benefit of extended cholecystectomy (open vs lap) should be evaluated between T1b and T2 (19 pts). Including T1a could lead to misinterpretation of the outcomes.
  • How are the survival results in un-matched cohort? OS and DFS

According to the sample size (about 100 pts) I suggest to abandon the idea of PSM but to describe extensively and thoroughly the study population and the comparison of open vs lap in the whole study population, or divided by T stage or other clinical and pathological characteristics (incidental?)

Author Response

Response to Reviewer 1 Comments

Point 1: Looking at the matched cohort I noted that there are 2 T1a and 19 T2 but no T1b. So, considering that for T1a GBC a simple cholecystectomy is recommended, the real benefit of extended cholecystectomy (open vs lap) should be evaluated between T1b and T2 (19 pts). Including T1a could lead to misinterpretation of the outcomes. According to the sample size (about 100 pts) I suggest to abandon the idea of PSM but to describe extensively and thoroughly the study population and the comparison of open vs lap in the whole study population, or divided by T stage or other clinical and pathological characteristics (incidental?).

Response 1: We thank the reviewer for this comment. Based on your suggestion, we have revised the text in the manuscript. We hope that this revised manuscript meets your expectations.

We used a PSM analysis to reduce variability between the open and laparoscopy groups.

Due to the small number of T1 patients, particularly in the Open group, we retitled the study "Comparison of the oncological outcomes of open versus laparoscopic surgery for T2 gallbladder cancer: A propensity-score-matched analysis" and modified the inclusion criteria as well as the statistical analysis.

Point 2: How are the survival results in un-matched cohort? OS and DFS

Response 2: We agree with the reviewer’s comment and have therefore revised the manuscript(page 3-4, figure 1-2)

“The median follow-up durations in the laparoscopic and open surgery groups were 21 and 48 months, respectively. The Kaplan–Meier curves for DFS and OS are shown in Figures 1 and 2, respectively. The two groups showed no significant differences in the 3-year DFS and OS rates (DFS: 65.0% vs. 66.7%, P = 0.721; cancer-specific OS: 78.0% vs. 82.4%, P = 0.782).”

Reviewer 2 Report

The authors compared outcomes of laparoscopic and open surgery for T1/T2 GBC under PSM. Although, this report seems interesting, it contains some critical points to be revised as is listed below.

Major points

  • The authors assessed outcome of T1 and T2 GBC but majority of the patients of T1a and T1b were listed in the Lap group. Considering this small number of T1 patients especially in Open group, this study should be modified as “Comparison of outcome of Lap vs. Open surgery for treatment of T2 GBC” by modifying inclusion criteria and also statistical analysis.

  • The authors described OS and DFS only for PSM cohort. They should describe OS and DFS of 114 patients before PSM.

  • The authors should describe the reason why they selected age, PS, T factor, type of surgical procedures and adjuvant chemotherapy for stratification factors of PSM. Because in Table 5, elevated CEA and lymph node metastases were significant in univariate and only the lymph metastases were the sole prognostic factor for survival.

Minor points

  • “T2b” of Table 3 could be “T1b”, the authors should revise table.

  • In the table 5, the authors set cutoff values of age, CEA, CA19-9 and tumor size without showing specific criteria. Please describe it in the statistical analysis section how did they decide.

Author Response

Response to Reviewer 2 Comments

Point 1: The authors assessed outcome of T1 and T2 GBC but majority of the patients of T1a and T1b were listed in the Lap group. Considering this small number of T1 patients especially in Open group, this study should be modified as “Comparison of outcome of Lap vs. Open surgery for treatment of T2 GBC” by modifying inclusion criteria and also statistical analysis.

Response 1: We thank the reviewer for this comment. Based on your suggestion, we have revised the text in the manuscript. We hope that this revised manuscript meets your expectations. Due to the small number of T1 patients, particularly in the Open group, we retitled the study "Comparison of the oncological outcomes of open versus laparoscopic surgery for T2 gallbladder cancer: A propensity-score-matched analysis" and modified the inclusion criteria as well as the statistical analysis.

Point 2: The authors described OS and DFS only for PSM cohort. They should describe OS and DFS of 114 patients before PSM.

Response 2: We thank the reviewer for indicating this point. We agree with the reviewer’s comment and have therefore revised the manuscript. (page 3-4, figure 1-2)

“The median follow-up durations in the laparoscopic and open surgery groups were 21 and 48 months, respectively. The Kaplan–Meier curves for DFS and OS are shown in Figures 1 and 2, respectively. The two groups showed no significant differences in the 3-year DFS and OS rates (DFS: 65.0% vs. 66.7%, P = 0.721; cancer-specific OS: 78.0% vs. 82.4%, P = 0.782).”

Point 3: The authors should describe the reason why they selected age, PS, T factor, type of surgical procedures and adjuvant chemotherapy for stratification factors of PSM. Because in Table 5, elevated CEA and lymph node metastases were significant in univariate and only the lymph metastases were the sole prognostic factor for survival.

Response 3: We agree with the reviewer’s comment and have therefore revised the manuscript.

“Many propensity models were tested with various covariates such as age, gender, preoperative American Society of Anesthesiologists (ASA) score, combined GB stone, pathologic T stage, pathologic N stage, simple cholecystectomy versus cholecystectomy with further resection, elevated carcinoembryonic antigen, elevated carbohydrate antigen 19-9, and adjuvant chemotherapy. After propensity score matching, we calculated the C-statistic and the standardized difference to get the best model. We chose the best-balanced PS matching model, which contained five covariates: age, preoperative ASA score, pathologic T stage, simple cholecystectomy versus cholecystectomy with further resection, and adjuvant chemotherapy. After matching, all covariates had reduced standardized differences and were well balanced between the two groups (C-statistic = .808).” (Page. 3)

Point 4: T2b” of Table 3 could be “T1b”, the authors should revise table.

Response 4: We thank the reviewer for this comment. Based on your suggestion, we have revised the table 3 in the manuscript. (Page. 5)

Point 5: In the table 5, the authors set cutoff values of age, CEA, CA19-9 and tumor size without showing specific criteria. Please describe it in the statistical analysis section how did they decide.

Response 5: We agree with the reviewer’s comment and have therefore revised the manuscript (page 3).

“CA19-9 and CEA were considered elevated if they were greater than 37 U/mL and greater than 5 ng/mL, respectively, according to the laboratory cut-off values often used in our center. Previous studies have routinely employed the cut-off age and tumor size of 60 years and greater than 1cm.”